# Deep Marching Tetrahedra: a Hybrid Representation for High-Resolution 3D Shape Synthesis

**Tianchang Shen** [1,2,3]    **Jun Gao**[1,2,3]    **Kangxue Yin** [1]

**Ming-Yu Liu** [1]    **Sanja Fidler**[1,2,3]

NVIDIA[1]    University of Toronto[2]    Vector Institute[3]

{frshen, jung, kangxuey, mingyul, sfidler}@nvidia.com

## Abstract

We introduce DMTET, a deep 3D conditional generative model that can synthesize high-resolution 3D shapes using simple user guides such as coarse voxels. It marries the merits of implicit and explicit 3D representations by leveraging a novel hybrid 3D representation. Compared to the current implicit approaches, which are trained to regress the signed distance values, DMTET directly optimizes for the reconstructed surface, which enables us to synthesize finer geometric details with fewer artifacts. Unlike deep 3D generative models that directly generate explicit representations such as meshes, our model can synthesize shapes with arbitrary topology. The core of DMTET includes a deformable tetrahedral grid that encodes a discretized signed distance function and a differentiable marching tetrahedra layer that converts the implicit signed distance representation to the explicit surface mesh representation. This combination allows joint optimization of the surface geometry and topology as well as generation of the hierarchy of subdivisions using reconstruction and adversarial losses defined explicitly on the surface mesh. Our approach significantly outperforms existing work on conditional shape synthesis from coarse voxel inputs, trained on a dataset of complex 3D animal shapes. Project page: https://nv-tlabs.github.io/DMTet/.

## 1 Introduction

Fields such as simulation, architecture, gaming, and film rely on high-quality 3D content with rich geometric details and complex topology. However, creating such content requires tremendous expert human effort. It takes a significant amount of development time to create each individual 3D asset. In contrast, creating rough 3D shapes with simple building blocks like voxels has been widely adopted. For example, Minecraft has been used by hundreds of millions of users for creating 3D content. Most of them are non-experts. Developing A.I. tools that enable regular people to upscale coarse, voxelized objects into high resolution, beautiful 3D shapes would bring us one step closer to democratizing high-quality 3D content creation. Similar tools can be envisioned for turning 3D scans of objects recorded by modern phones into high-quality forms. Our work aspires to create such capabilities.

A powerful 3D representation is a critical component of a learning-based 3D content creation framework. A good 3D representation for high-quality reconstruction and synthesis should capture local geometric details and represent objects with arbitrary topology while also being memory and computationally efficient for fast inference in interactive applications.

Recently, neural implicit representations [8, 39, 42, 51], which use a neural network to implicitly represent a shape via a signed distance field (SDF) or an occupancy field (OF), have emerged as an effective 3D representation. Neural implicits have the benefit of representing complex geometry

35th Conference on Neural Information Processing Systems (NeurIPS 2021).

and topology, not limited to a predefined resolution. The success of these methods has been shown in shape compression [49, 13, 51], single-image shape generation [47, 60, 48], and point cloud reconstruction [57]. However, most of the current implicit approaches are trained by regressing to SDF or OF values and cannot utilize an explicit supervision on the target surface, which imposes useful constraints for training. To mitigate this issue, several works [45, 31] proposed to utilize iso-surfacing techniques such as the Marching Cubes (MC) algorithm to extract a surface mesh from the implicit representation, which, however, is computationally expensive.

In this work, we introduce DMTET, a deep 3D conditional generative model for high-resolution 3D shape synthesis from user guides in the form of coarse voxels. In the heart of DMTET is a new differentiable shape representation that marries implicit and explicit 3D representations. In contrast to deep implicit approaches optimized for predicting sign distance (or occupancy) values, our model employs additional supervision on the surface, which empirically renders higher quality shapes with finer geometric details. Compared to methods that learn to directly generate explicit representations, such as meshes [54], by committing to a preset topology, our DMTET can produce shapes with arbitrary topology. Specifically, DMTET predicts the underlying surface parameterized by an implicit function encoded via a deformable tetrahedral grid. The underlying surface is converted into an explicit mesh with a Marching Tetrahedra (MT) algorithm, which we show is differentiable and more performant than the Marching Cubes. DMTET maintains efficiency by learning to adapt the grid resolution by deforming and selectively subdividing tetrahedra. This has the effect of spending computation only on the relevant regions in space. We achieve further gains in the overall quality of the output shape with learned surface subdivision. Our DMTET is end-to-end differentiable, allowing the network to jointly optimize the geometry and topology of the surface, as well as the hierarchy of subdivisions using a loss function defined explicitly on the surface mesh.

We demonstrate our DMTET on two challenging tasks: 3D shape synthesis from coarse voxel inputs and point cloud 3D reconstruction. We outperform existing state-of-the-art methods by a significant margin while being 10 times faster than alternative implicit representation-based methods at inference time. In summary, we make the following technical contributions:

1. We show that using Marching Tetrahedra (MT) as a differentiable iso-surfacing layer allows topological change for the underlying shape represented by a implicit field, in contrast to the analysis in prior works [31, 45].

2. We incorporate MT in a DL framework and introduce DMTET, a hybrid representation that combines implicit and explicit surface representations. We demonstrate that the additional supervision (e.g. chamfer distance, adversarial loss) defined directly on the extracted surface from implicit field improves the shape synthesis quality.

3. We introduce a coarse-to-fine optimization strategy that scales DMTET to high resolution during training. We thus achieves better reconstruction quality than state-of-the-art methods on challenging 3D shape synthesis tasks, while requiring a lower computation cost.

## 2 Related Work

We review the related work on learning-based 3D synthesis methods based on their 3D representations.

**Voxel-based Methods**   Early work [59, 10, 38] represented 3D shapes as voxels, which store the coarse occupancy (inside/outside) values on a regular grid, which makes powerful convolutional neural networks native and renders impressive results on 3D reconstruction and synthesis [12, 11, 58, 2]. For high-resolution shape synthesis, DECOR-GAN [6] transfers geometric details from a high-resolution shape represented in voxel to a low-resolution shape by utilizing a discriminator defined on 3D patches of the voxel grid. However, the computational and memory costs grow cubically as the resolution increases, prohibiting the reconstruction of fine geometric details and smooth curves. One common way to address this limitation is building hierarchical structures such as octrees [46, 52, 55, 56, 24, 52], which adapt the grid resolution locally based on the underlying shape. In this paper, we adopt a hierarchical deformable tetrahedral grid to utilize the resolution better. Unlike octree-based shape synthesis, our network learns grid deformation and subdivision jointly to better represent the surface without relying on explicit supervision from a pre-computed hierarchy.

**Deep Implicit Fields (DIFs)**   represent a 3D shape as a zero level set of a continuous function parameterized by a neural network [39, 44, 17, 40]. This formulation can represent arbitrary typology and has infinite resolution. DIF-based shape synthesis approaches have demonstrated strong

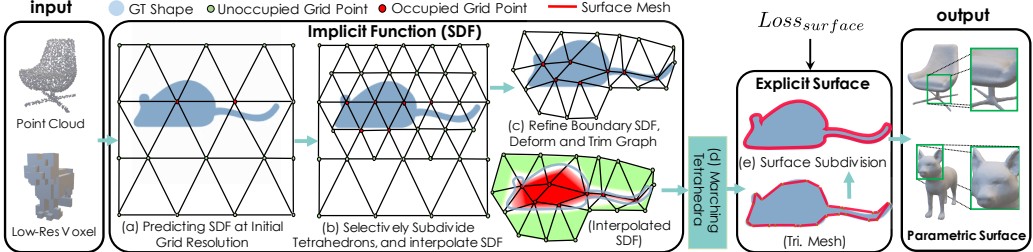

Figure 1: **DMTET** reconstructs the shape implicitly in a coarse-to-fine manner by predicting the SDF defined on a deformable tetrahedral grid. It then converts the SDF to a surface mesh by a differentiable Marching Tetrahedra layer. **DMTET is trained by optimizing the objective function defined on the final surface.**

performance in many applications, including single view 3D reconstruction [60, 30, 47, 48], shape manipulation, and synthesis [26, 21, 28, 14, 1, 9]. However, as these approaches are trained by minimizing the reconstruction loss of function values at a set of sampled 3D locations (a rough proxy of the surface), they tend to render artifacts when synthesizing fine details. Furthermore, if one desires a mesh to be extracted from a DIF, an expensive iso-surfacing step based on Marching Cubes [36] or Marching Tetrahedra [15] is required. Due to the computational burden, iso-surfacing is often done on a smaller resolution, hence prone to quantization errors. Lei et al. [29] proposes an analytic meshing solution to reduce the error, but is only applicable to DIFs parametrized by MLPs with ReLU activation. Our representation scales to high resolution and does not require additional modification to the backward pass for training end-to-end. DMTET can represent arbitrary typology, and is trained via direct supervision on the generated surface. Recent works [1, 9] learn to regress unsigned distance to triangle soup or point cloud. However, their iso-surfacing formulation is not differentiable in contrast to DMTET.

**Surface-based Methods** directly predict triangular meshes and have achieved impressive results for reconstructing and synthesizing simpler shapes [54, 23, 5, 41, 7]. Typically, they predefined the topology of the shape, e.g. equivalent to a sphere [54, 5, 25], or a union of primitives [43, 53, 19] or a set of segmented parts [61, 62, 50]. As a result, they can not model a distribution of shapes with complex topology variations. Recently, DefTet [18] represents a mesh with a deformable tetrahedral grid where the grid vertex coordinates and the occupancy values are learned. However, similar to voxel-based methods, the computational costf increases cubically with the grid resolution. Furthermore, as the occupancy loss for supervising topology learning and the surface loss for supervising geometry learning do not support joint training, it tends to generate suboptimal results. In contrast, our method is able to synthesize high-resolution 3D shapes, not shown in previous work.

## 3 Deep Marching Tetrahedra

We now introduce our DMTET for synthesizing high-quality 3D objects. The schematic illustration is provided in Fig. 1. Our model relies on a new, hybrid 3D representation specifically designed for high-resolution reconstruction and synthesis, which we describe in Sec. 3.1. In Sec. 3.2, we describe the neural network architecture of DMTET that predicts the shape representation from inputs such as coarse voxels. We provide the training objectives in Sec. 3.3.

### 3.1 3D Representation

We represent a shape using a sign distance field (SDF) encoded with a deformable tetrahedral grid, adopted from DefTet [18, 20]. The grid fully tetrahedralizes a unit cube, where each cell in the volume is a tetrahedron with 4 vertices and faces. The key aspect of this representation is that the grid vertices can deform to represent the geometry of the shape more efficiently. While the original DefTet encoded occupancy defined on each tetrahedron, we here encode signed distance values defined on the vertices of the grid and represent the underlying surface implicitly (Sec. 3.1.1). The use of signed distance values, instead of occupancy values, provides more flexibility in representing the underlying surface. For greater representation power while keeping memory and computation manageable, we further selectively subdivide the tetrahedra around the predicted surface (Sec. 3.1.2). We convert the signed distance-based implicit representation into a triangular mesh using a marching tetrahedra layer, which we discuss in Sec. 3.1.3. The final mesh is further converted into a parameterized surface with a differentiable surface subdivision module, described in Sec. 3.1.4.

### 3.1.1 Deformable Tetrahedral Mesh as an Approximation of an Implicit Function

We adopt and extend the deformable tetrahedral grid introduced in Gao et al. [18], which we denote with $(V_T, T)$, where $V_T$ are the vertices in the tetrahedral grid $T$. Following the notation in [18], each tetrahedron $T_k \in T$ is represented with four vertices $\{v_{a_k}, v_{b_k}, v_{c_k}, v_{d_k}\}$, with $k \in \{1, \ldots, K\}$, where $K$ is the total number of tetrahedra and $v_{i_k} \in V_T$.

We represent the sign distance field by interpolating SDF values defined on the vertices of the grid. Specifically, we denote the SDF value in vertex $v_i \in V_T$ as $s(v_i)$. SDF values for the points that lie inside the tetrahedron follow a barycentric interpolation of the SDF values of the four vertices that encapsulates the point.

### 3.1.2 Volume Subdivision

We represent shape in a coarse to fine manner for efficiency. We determine the *surface tetrahedra* $T_{surf}$ by checking whether a tetrahedron has vertices with different SDF signs – indicating that it intersects the surface encoded by the SDF. We subdivide $T_{surf}$ as well as their immediate neighbors and increase resolution by adding the mid point to each edge. We compute SDF values of the new vertices by averaging the SDF values on the edge (Fig. 2).

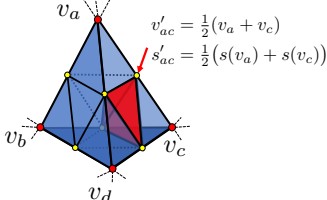

Figure 2: **Volume Subdivision:** Each surface tet.(blue) is divided into 8 tet.(red) by adding midpoints.

### 3.1.3 Marching Tetrahedra for converting between an Implicit and Explicit Representation

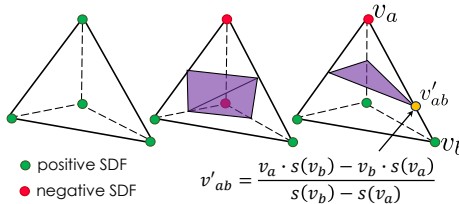

Figure 3: Three unique surface configurations in MT. Vertex color indicates the sign of signed distance value. Notice that flipping the signs of all vertices will result in the same surface configuration. Position of the vertex is linearly interpolated along the edges with sign change.

We use the Marching Tetrahedra [15] algorithm to convert the encoded SDF into an explicit triangular mesh. Given the SDF values $\{s(v_a), s(v_b), s(v_c), s(v_d)\}$ of the vertices of a tetrahedron, MT determines the surface typology inside the tetrahedron based on the signs of $s(v)$, which is illustrated in Fig. 3. The total number of configurations is $2^4 = 16$, which falls into 3 unique cases after considering rotation symmetry. Once the surface typology inside the tetrahedron is identified, the vertex location of the iso-surface is computed at the zero crossings of the linear interpolation along tetrahedron's edges, as shown in Fig. 3.

Prior works [45, 31] argue that the singularity in this formulation, i.e. when $s(v_a) = s(v_b)$, prevents the change of surface typology (sign change of $s(v_a)$) during training. However, we find that, in practise, the equation is only evaluated when $\text{sign}(s(v_a)) \neq \text{sign}(s(v_b))$. Thus, during training, the singularity never happens and the gradient from a loss defined on the extracted iso-surface (Sec. 3.3), can be back-propagated to both vertex positions and SDF values via the chain rule. A more detailed analysis is in the Appendix.

### 3.1.4 Surface Subdivision

Having a surface mesh as output allows us to further increase the representation power and the visual quality of the shapes with a differentiable surface subdivision module. We follow the scheme of the Loop Subdivision method [35], but instead of using a fixed set of parameters for subdivision, we make these parameters learnable in DMTET. Specifically, learnable parameters include the positions of each mesh vertex $v'_i$, as well as $\alpha_i$ which controls the generated surface via weighting the smoothness of neighbouring vertices. Note that different from Liu et al. [33], we only predict the per-vertex parameter at the beginning and carry it over to subsequent subdivision iterations to attain a lower computational cost. We provide more details in Appendix.

### 3.2 DMTET: 3D Deep Conditional Generative Model

Our DMTET is a neural network that utilizes our proposed 3D representation and aims to output a high resolution 3D mesh $M$ from input $x$ (a point cloud or a coarse voxelized shape). We describe the architecture (Fig. 4) of the generator for each module of our 3D representation in Sec. 3.2.1, with the architecture of the discriminator presented in Sec. 3.2.2. Further details are in Appendix.

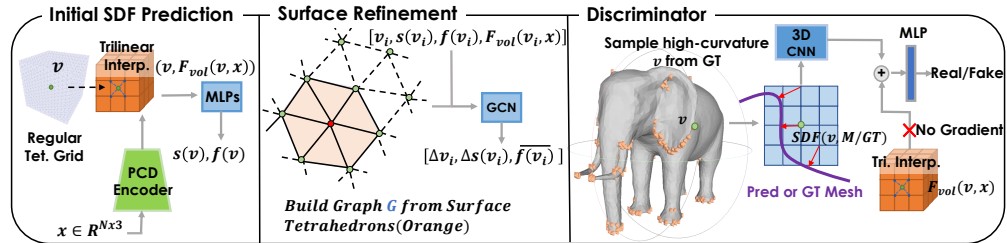

Figure 4: Our generator and discriminator architectures. The generator is composed of two parts—one utilizes MLP to generate the initial predictions for all grid vertices and the other uses GCN to refine the surface.

### 3.2.1 3D Generator

**Input Encoder** We use PVCNN [34] as an input encoder to extract a 3D feature volume $F_{vol}(x)$ from a point cloud. When the input is a coarse voxelized shape, we sample points on its surface. We compute a feature vector $F_{vol}(v, x)$ for a grid vertex $v \in \mathbb{R}^3$ via trilinear interpolation.

**Initial Prediction of SDF** We predict the SDF value for each vertex in the initial deformable tetrahedral grid using a fully-connected network $s(v) = MLP(F_{vol}(v, x), v)$. The fully-connected network additionally outputs a feature vector $f(v)$, which is used for the surface refinement in the volume subdivision stage.

**Surface Refinement with Volume Subdivision** After obtaining the initial SDF, we iteratively refine the surface and subdivide the tetrahedral grid. We first identify surface tetrahedra $T_{surf}$ based on the current $s(v)$ value. We then build a graph $G = (V_{surf}, E_{surf})$, where $V_{surf}, E_{surf}$ correspond to the vertices and edges in $T_{surf}$. We then predict the position offsets $\Delta v_i$ and SDF residual values $\Delta s(v_i)$ for each vertex $i$ in $V_{surf}$ using a Graph Convolutional Network [32] (GCN):

$$f'_{v_i} = \text{concat}(v_i, s(v_i), F_{vol}(v_i, x), f(v_i)), \quad (1)$$

$$(\Delta v_i, \Delta s(v_i), \overline{f(v_i)})_{i=1, \cdots N_{surf}} = \text{GCN}\big((f'_{v_i})_{i=1, \cdots N_{surf}}, G\big), \quad (2)$$

where $N_{surf}$ is the total number of vertices in $V_{surf}$ and $\overline{f(v_i)}$ is the updated per-vertex feature. The vertex position and the SDF value for vertex $v_i$ are updated as $v'_i = v_i + \Delta v_i$ and $s(v'_i) = s(v_i) + \Delta s(v_i)$. This refinement step can potentially flip the sign of the SDF values to refine the local typology, and also move the vertices thus improving the local geometry.

After surface refinement, we perform the volume subdivision step followed by an additional surface refinement step. In particular, we re-identify $T_{surf}$ and subdivide $T_{surf}$ and their immediate neighbors. We drop the unsubdivided tetrahedra from the full tetrahedral grid in both steps, which saves memory and computation, as the size of the $T_{surf}$ is proportional to the surface area of the object, and scales up quadratically rather than cubically as the grid resolution increases.

Note that the SDF values and positions of the vertices are inherited from the level before subdivision, thus, the loss computed at the final surface can back-propagate to all vertices from all levels. Therefore, our DMTET automatically learns to subdivide the tetrahedra and does not need an additional loss term in the intermediate steps to supervise the learning of the octree hierarchy as in the prior work [52].

**Learnable Surface Subdivision** After extracting the surface mesh using MT, we can further apply learnable surface subdivision. Specifically, we build a new graph on the extracted mesh, and use GCN to predict the updated position of each vertex $v'_i$, and $\alpha_i$ for Loop Subdivision. This step removes the quantization errors and mitigates the approximation errors from the classic Loop Subdivision by adjusting $\alpha_i$, which are fixed in the classic method.

### 3.2.2 3D Discriminator

We apply a 3D discriminator $D$ on the final surface predicted from the generator. We empirically find that using a 3D CNN from DECOR-GAN [6] as the discriminator on the signed distance field that is computed from the predicted mesh is effective to capture the local details. Specifically, we first randomly select a high-curvature vertex $v$ from the target mesh and compute the ground truth signed distance field $S_{real} \in \mathbb{R}^{N \times N \times N}$ at a voxelized region around $v$. Similarly, we compute the signed distance field of the predicted surface mesh $M$ at the same location to obtain $S_{pred} \in \mathbb{R}^{N \times N \times N}$. Note that $S_{pred}$ is an analytical function of the mesh $M$, and thus the gradient to $S_{pred}$ can back-propagate to the vertex positions in $M$. We feed $S_{real}$ or $S_{pred}$ into the discriminator, along with the feature vector $F_{vol}(v, x)$ in position $v$. The discriminator then predicts the probability indicating whether the input comes from the real or generated shapes.

### 3.3 Loss Function

DMTET is end-to-end trainable. We supervise all modules to minimize the error defined on the final predicted mesh $M$. Our loss function contains three different terms: a surface alignment loss to encourage the alignment with ground truth surface, an adversarial loss to improve realism of the generated shape, and regularizations to regularize the behavior of SDF and vertex deformations.

**Surface Alignment loss**  We sample a set of points $P_{gt}$ from the surface of the ground truth mesh $M_{gt}$. Similarly, we also sample a set of points from $M_{pred}$ to obtain $P_{pred}$, and minimize the L2 Chamfer Distance and the normal consistency loss between $P_{gt}$ and $P_{pred}$ :

$$L_{cd} = \sum_{p \in P_{pred}} \min_{q \in P_{gt}} ||p - q||_2 + \sum_{q \in P_{gt}} \min_{p \in P_{pred}} ||q - p||_2, L_{normal} = \sum_{p \in P_{pred}} (1 - |\vec{\mathbf{n}}_p \cdot \vec{\mathbf{n}}_{\hat{q}}|), \quad (3)$$

where $\hat{q}$ is the point that corresponds to $p$ when computing the Chamfer Distance, and $\vec{\mathbf{n}}_p, \vec{\mathbf{n}}_{\hat{q}}$ denotes the normal direction at point $p, \hat{q}$.

**Adversarial Loss**  We use the adversarial loss proposed in LSGAN [37]:

$$L_{D} = \frac{1}{2}[(D(M_{gt}) - 1)^2 + D(M_{pred})^2], \; L_{G} = \frac{1}{2}[(D(M_{pred}) - 1)^2]. \quad (4)$$

**Regularizations**  The above loss functions operate on the extracted surface, thus, only the vertices that are close to the iso-surface in the tetrahedral grid receive gradients, while the other vertices do not. Moreover, the surface losses do not provide information about what is inside/outside, since flipping the SDF sign of all vertices in a tetrahedron would result in the same surface being extracted by MT. This may lead to disconnected components during training. To alleviate this issue, we add a SDF loss to regularize SDF values:

$$L_{SDF} = \sum_{v_i \in V_T} |s(v_i) - SDF(v_i, M_{gt})|^2, \quad (5)$$

where $SDF(v_i, M_{gt})$ denotes the SDF value of point $v_i$ to the mesh $M_{gt}$. In addition, we apply the $L_2$ regularization loss on the predicted vertex deformations to avoid artifacts: $L_{def} = \sum_{v_i \in V_T} ||\Delta v_i||_2$.

The final loss is a weighted sum of all five loss terms:

$$L = \lambda_{cd} L_{cd} + \lambda_{normal} L_{normal} + \lambda_{G} L_{G} + \lambda_{SDF} L_{SDF} + \lambda_{def} L_{def}, \quad (6)$$

where $\lambda_{cd}, \lambda_{normal}, \lambda_{G}, \lambda_{SDF}, \lambda_{def}$ are hyperparameters (provided in the Supplement).

## 4 Experiments

We first evaluate DMTET in the challenging application of generating high-quality animal shapes from coarse voxels. We further evaluate DMTET in reconstructing 3D shapes from noisy point clouds on ShapeNet by comparing to existing state-of-the-art methods.

### 4.1 3D Shape Synthesis from Coarse Voxels

**Experimental Settings**  We collected 1562 animal models from the TurboSquid website[1]. These models have a wide range of diversity, ranging from cats, dogs, bears, giraffes, to rhinoceros, goats, etc. We provide visualizations in Supplement. Among 1562 shapes, we randomly select 1120 shapes for training, and the remaining 442 shapes for testing. We follow the pipeline in Kaolin [27] to convert shapes to watertight meshes. To prepare the input to the network, we first voxelize the mesh into the resolution of $16^3$, and then sample 3000 points from the surface after applying marching cubes to the $16^3$ voxel grid. Note that this preprocessing is agnostic to the representation of the input coarse shape, allowing us to evaluate on different resolution voxels, or even meshes.

We compare our model with the official implementation of ConvOnet [44], which achieved SOTA performance on voxel upsampling. We also compare to DECOR-GAN [6], which obtained impressive results on transferring styles from a high-resolution voxel shape to a low-resolution voxel. Note that the original setting of DECOR-GAN is different from ours. For a fair comparison, we use all 1120 training shapes as the high-resolution style shapes during training, and retrieve the closet training shape to the test shape as the style shape during inference, which we refer as DECOR-Retv. We also compare against a randomly selected style shape as reference, denoted as DECOR-Rand.

---

[1]https://www.turbosquid.com, we obtain consent via an agreement with TurboSquid, and following license at https://blog.turbosquid.com/turbosquid-3d-model-license/

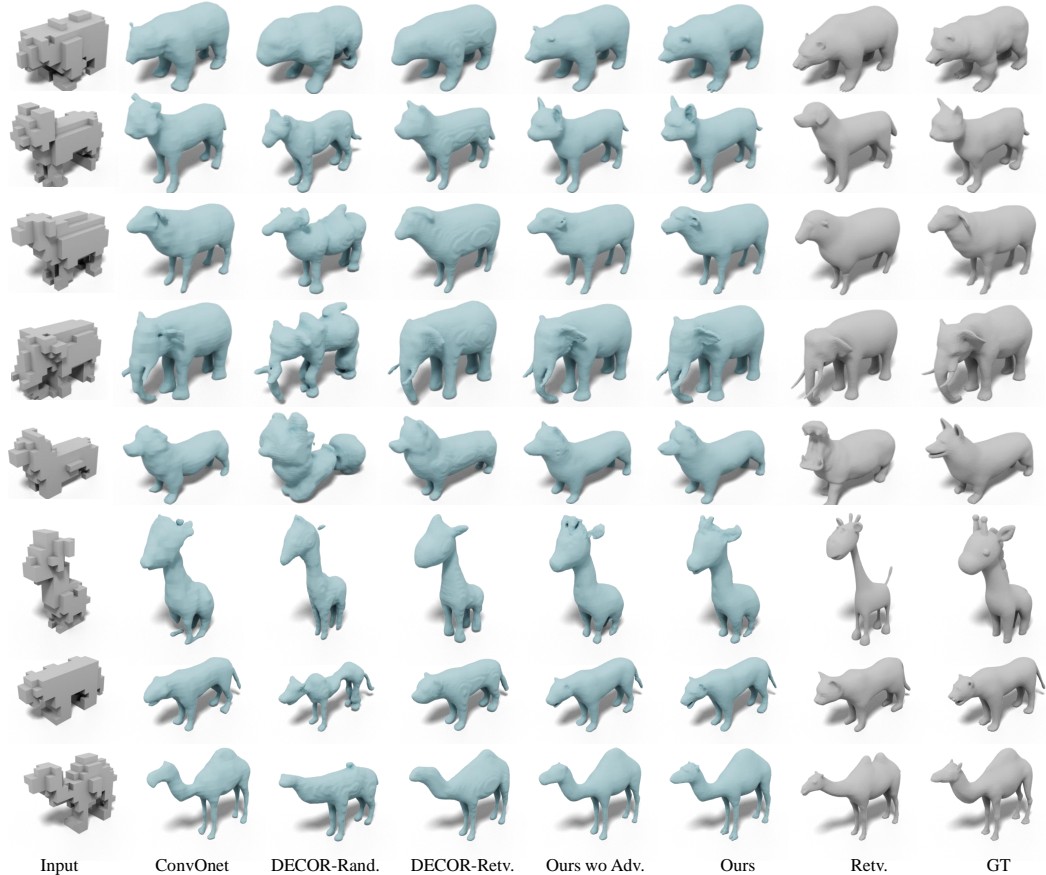

| Input | ConvOnet | DECOR-Rand. | DECOR-Retv. | Ours wo Adv. | Ours | Retv. | GT |

Figure 5: **Qualitative results on 3D shapes Synthesis from Coarse Voxels**. Comparing with all baselines, our method reconstructs shapes with much higher quality. Adding GAN further improves the realism of the generated shape. We also show the retrieved shapes from the training set in the second last column.

**Metrics**   We evaluate L2 and L1 Chamfer Distance, as well as normal consistency score to assess how well the methods reconstruct the corresponding high-resolution shape following [44]. We also report Light Field Distance [4] (LFD) which measures the visual similarity in 2D rendered views. In addition, we evaluate Cls score following [6]. Specifically, we render the predicted 3D shapes and train a patch-based image classifier to distinguish whether images are from the renderings of real or generated shapes. The mean classification accuracy of the trained classifier is reported as Cls (lower is better). More details are in the Supplement.

**Experimental Results**   We provide quantitative results in Table 1 with qualitative examples in Fig. 5. Our DMTET achieves significant improvements over all baselines in terms of all metrics. Compared to both ConvOnet [44] and DECOR-GAN [6], our DMTET reconstructs shapes with better quality when training without adversarial loss (5th column in Fig. 5). Further geometric details, including nails, ears, eyes, mouths, etc, are captured when trained with the adversarial loss (6th column in Fig. 5), significantly improving the realism and visual quality of the generated shape. To demonstrate the generalization ability of our DMTET, we collect human-created low-resolution

Figure 6: Qualitative Results of synthesizing high-resolution shapes from coarse voxels collected online.

voxels from Turbosquid (shapes unseen in training). We provide qualitative results in Fig. 6. Despite the fact that these human-created shapes have noticeable differences with our coarse voxels used in training, e.g., different ratios of body parts compared with our training shapes (larger head, thinner legs, longer necks), our model faithfully generates high-quality 3D details conditioned on each coarse voxel – an exciting result.

| | L2 Chamfer ↓ | L1 Chamfer ↓ | Norm. Cons. ↑ | LFD ↓ | Cls ↓ |
|---|---|---|---|---|---|
| ConvOnet [44] | 0.83 | 2.41 | 0.901 | 3220 | 0.63 |
| DECOR [6]-Retv. | 1.32 | 3.81 | 0.876 | 3689 | 0.66 |
| DECOR [6]-Rand. | 2.38 | 6.85 | 0.797 | 5338 | 0.67 |
| DMTET wo Adv. | 0.76 | 2.20 | **0.916** | 2846 | 0.58 |
| DMTET | **0.75** | **2.19** | 0.918 | **2823** | **0.54** |

Table 1: **Super Resolution of Animal Shapes**: DMTET significantly outperforms all baselines in all metrics.

**User Studies**   We conduct user studies via Amazon Machanical Turk (AMT) to further evaluate the performance of all methods. In particular, we present two shapes that are predicted from two different models to the AMT workers and ask them to evaluate which one is a better looking shape and which one features more realistic details. Detailed experimental settings are provided in the Supplement. We compare DMTET against ConvONet [44], DECOR [6]-Retv, as well as DMTET without adversarial loss (w.o. Adv.). Quantitative results are reported in Table 2. Human judges agree that the shapes generated from our model have better details, compared to all baselines, in a vast majority of the cases. Ablations on using adversarial loss demonstrate the effectiveness of generating higher quality geometry using a discriminator during training.

**Ablation Studies**   To evaluate the effectiveness of our volume subdivision and surface subdivision modules, we ablate by sequentially introducing them to the base model (we refer as $DMTET_B$) which we train on 100-resolution uniform tetrahedral grid without both volume and surface subdivision modules and adversarial loss. We conduct user studies to evaluate the improvement after each step using the protocol described

| | ConvONet [44] | DECOR [6]-Retv. | DMTET wo Adv. |
|---|---|---|---|
| Baseline wins | 5% / 5% | 26% / 17% | 29% / 25% |
| DMTET wins | 95% / 95% | 74% / 83% | 71% / 75% |

Table 2: User Study on 3D Shape Synthesis from Coarse voxels. In each cell, we report percentages of shapes for which the users agree are better looking (left) or have better details (right).

in the above paragraph. We first reduce the initial resolution to 70 and employ volume subdivision to support higher output resolution (we refer this model as $DMTET_V$) and compare with $DMTET_B$. Predictions by $DMTET_V$ wins 78% of cases over $DMTET_B$ for better looking, and 61% of cases for realistic details, showing that the volume subdivision module is effective in synthesizing shape details. We then add surface subdivision on top of the $DMTET_V$ and compare with it. The new model wins 62% of cases over $DMTET_V$ for better looking, and 62% of cases for realistic details as well, demonstrating the effect of surface subdivision module in enhancing the shape details.

## 4.2   Point Cloud 3D Reconstruction

**Experimental Settings**   We follow the setting from DefTet [18], and use all 13 categories in ShapeNet [3] core data[2], which we pre-process using Kaolin [27] to watertight meshes. We sample 5000 points for each shape and add Gaussian noise with zero mean of standard deviation 0.005. For quantitative evaluation, we report the L1 Chamfer Distance in the main paper, and refer readers to the Supplement for results in other metrics (3D IoU, L2 Chamfer Distance and F1 score). We additionally report average inference time on the same Nvidia V100 GPU.

We compare DMTET against state-of-the-art 3D reconstruction approaches using different representations: voxels [10], deforming a mesh with a fixed template [54], deforming a mesh generated from a volumetric representation [22], DefTet [18], and implicit functions [44]. For a fair comparison, we use the same point cloud encoder for all the methods, and adopt the decoders in the original papers to generate shapes in different representations. We also remove the adversarial loss in this application, since baselines also do not have it. We further compare with oracle performance of MC/MT where the ground truth SDF is utilized to extract iso-surface using MC/MT.

**Experimental Results**   Quantitative results are summarized in Table 3, with a few qualitative examples shown in Fig. 7. Compared to DMC [31], which also predicts the SDF values and supervises with a surface loss, DMTET achieves much better reconstruction quality since training using the marching tetrahedra layer is more efficient than calculating an expectation over all possible configurations within one grid cell as done in DMC [31]. Compared to a method that deforms a fixed template (sphere) [54], we reconstruct shapes with different topologies, achieving more faithful results compared to the ground truth shape. When compared with other explicit surface representations that also support different topology [18, 22], our method achieves higher quality results for local

---
[2]The ShapeNet license is explained at https://shapenet.org/terms

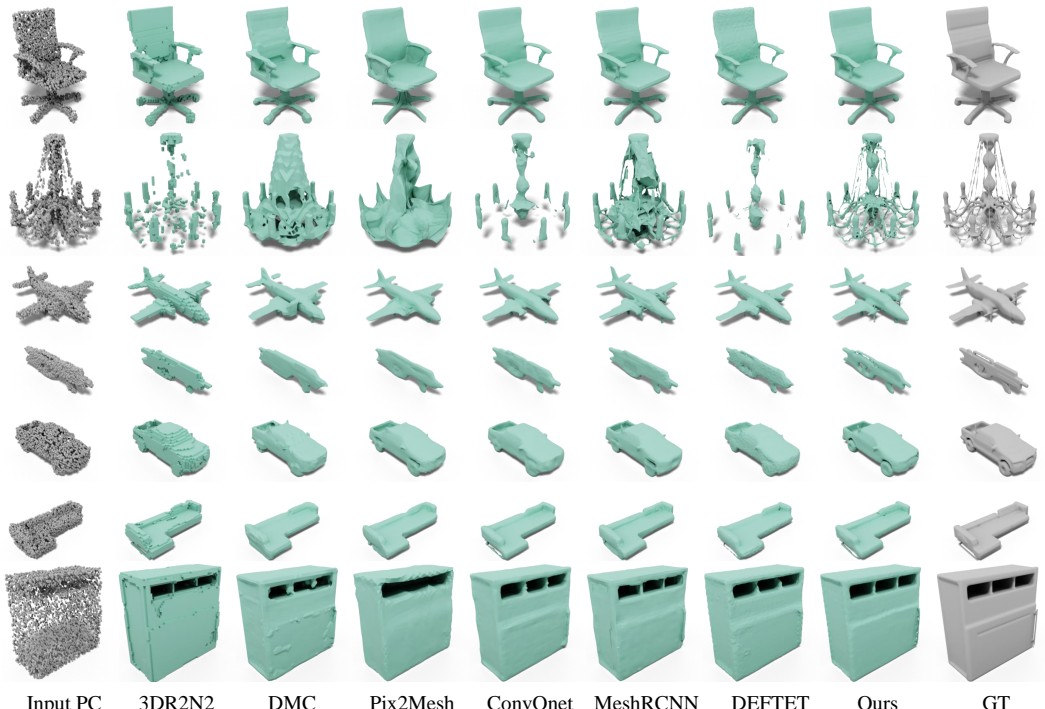

| | Input PC | 3DR2N2 | DMC | Pix2Mesh | ConvOnet | MeshRCNN | DEFTET | Ours | GT |

Figure 7: **Qualitative results on 3D Reconstruction from Point Clouds:** Our model reconstructs shapes with more geometric details compared to baselines.

| Category | Airplane | Bench | Dresser | Car | Chair | Display | Lamp | Speaker | Rifle | Sofa | Table | Phone | Vessel | **Mean↓** | **Time**(ms)↓ |
|---|---|---|---|---|---|---|---|---|---|---|---|---|---|---|---|
| 3D-R2N2 [10] | 1.48 | 1.59 | 1.64 | 1.62 | 1.70 | 1.66 | 1.74 | 1.74 | 1.37 | 1.60 | 1.78 | 1.55 | 1.51 | 1.61 | 174 |
| DMC [31] | 1.57 | 1.47 | 1.29 | 1.67 | 1.44 | 1.25 | 2.15 | 1.49 | 1.45 | 1.19 | 1.33 | 0.88 | 1.70 | 1.45 | 349 |
| Pixel2mesh [54] | 0.98 | 1.28 | 1.44 | 1.19 | 1.91 | 1.25 | 2.07 | 1.61 | 0.91 | 1.15 | 1.82 | 0.83 | 1.12 | 1.35 | **30** |
| ConvOnet [44] | 0.82 | 0.95 | 0.96 | 1.12 | 1.03 | 0.93 | 1.22 | 1.12 | 0.79 | 0.91 | 0.94 | 0.67 | 0.99 | 0.95 | 866 |
| MeshRCNN [22] | 0.88 | 1.01 | 1.05 | 1.14 | 1.10 | 0.99 | 1.20 | 1.21 | 0.83 | 0.96 | 1.00 | 0.71 | 1.03 | 1.01 | 228 |
| DEFTET [18] | 0.85 | 0.94 | 0.97 | 1.13 | 1.04 | 0.92 | 1.28 | 1.17 | 0.85 | 0.90 | 0.93 | 0.65 | 0.99 | 0.97 | 61 |
| DMTET wo (Def, Vol., Surf.) | 0.82 | 0.96 | 0.94 | 0.98 | 0.99 | 0.90 | 1.04 | 1.03 | 0.80 | 0.86 | 0.93 | 0.65 | 0.89 | 0.91 | 52 |
| DMTET wo (Vol., Surf.) | 0.69 | 0.82 | 0.88 | 0.92 | 0.92 | 0.82 | 0.89 | 0.97 | 0.65 | 0.81 | 0.84 | 0.61 | 0.80 | 0.81 | 52 |
| DMTET wo Vol. | 0.65 | 0.78 | 0.84 | 0.89 | 0.89 | 0.79 | 0.86 | 0.95 | 0.61 | 0.78 | 0.79 | 0.60 | 0.78 | 0.79 | 67 |
| DMTET wo Surf. | 0.63 | 0.77 | 0.84 | 0.88 | **0.88** | 0.79 | **0.84** | **0.94** | 0.60 | 0.78 | 0.79 | 0.59 | **0.76** | 0.78 | 108 |
| DMTET | **0.62** | **0.76** | **0.83** | **0.87** | **0.88** | **0.78** | **0.84** | **0.94** | **0.59** | **0.77** | **0.78** | **0.57** | **0.76** | **0.77** | 129 |

Table 3: Quantitative Results on **Point Cloud Reconstruction** (Chamfer L1). Note that all the networks in the baselines are not designed for this task, and thus we use the same encoder and their decoder for a fair comparison. We also ablate ourselves by operating on fixed grid (DMTET wo (Def, Vol., Surf.)), removing volume subdivision (DMTET wo Vol.), or surface subdivision (DMTET wo Surf.), or the both (DMTET wo (Vol., Surf.)).

geometry, benefiting from the fact that the typology is jointly optimized with the geometry, whereas it is separately supervised by an occupancy loss in [18, 22]. Compared to a neural implicit method [44], we generate higher quality shapes with less artifacts, while running significantly faster at inference. Finally, compared to a voxel-based method [10] at the same resolution, our method recovers more geometric details, benefiting from the predicted vertex deformations as well as the surface loss.

### 4.2.1 Analysis

We investigate how each component in our representation affects the performance and reconstruction quality.

**Comparisons with Oracle Performance of MC/MT** We first demonstrate the effect of learning on explicit surface via MT. We compare with the oracle performance of extracting the iso-surface with MT/MC from the ground truth signed distance fields on the Chair test set in ShapeNet, which contains diverse high-quality details. Specifically, for MC/MT, we first compute the discretized SDF at different grid resolutions, and compare the extracted surface to the ground truth surface.

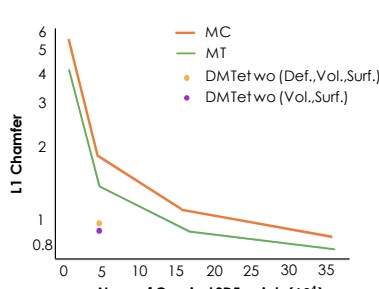

Figure 8: Comparing our DMTET with oracle performance of MC and MT.

As shown in Fig. 8, MT consistently outperforms MC when querying the same number of points. We found the staggered grids pattern in tetrahedral grid [16, 18] better captures thin structures at a limited

resolution (Fig. 9). This makes MT a better choice for efficiency reasons. The usage of tetrahedral mesh in DMTET follows this motivation. Without deforming the grid, DMTET outperforms the oracle performance of MT by a large margin when querying the same number of points, although DMTET predicts the surface from noisy point cloud. This demonstrates that directly optimizing the reconstructed surface can mitigate the discretization errors imposed by MT to a large extent.

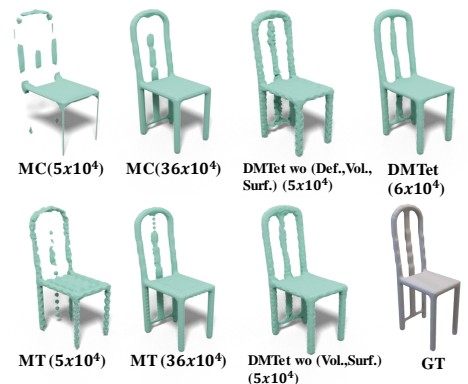

Figure 9: We compare trained DMTET to oracle performance of MT and MC. Number in bracket indicates number of SDF points queried.

**Ablation Studies** We further provide ablation studies on the entire ShapeNet test set, which is summarized in Tab. 3. We first compare the version where we only predict SDF values without learning to deform the vertices and volume/surface subdivision with the version that predicts both SDF and the deformation. Predicting deformation along with SDF is significantly more performant, since vertex movements allow for a better reconstruction of the underlying surface. This is especially true for categories with thin structures (e.g. lamp) where the grid vertices are desired to align with them. We further ablate the use of volume subdivision and surface subdivision. We show that each component provides an improvement. In particular, volume subdivision has a significant improvement for object categories with fine-grained structural details, such as airplane and lamp, which require higher grid resolutions to model the occupancy change. Surface subdivision generates shapes with a parametric surface, avoiding the quantization errors in the planar faces and produces more visually pleasing results.

## 5 Conclusion

In this paper, we introduced a deep 3D conditional generative model that can synthesize high-resolution 3D shapes using simple user guides such as coarse voxels. Our DMTET features a novel 3D representation that marries implicit and explicit representations by leveraging the advantages of both. We experimentally show that our approach synthesizes significantly higher quality shapes with better geometric details than existing methods, confirmed by quantitative metrics and an extensive user study. By showcasing the ability to upscale coarse voxels such as Minecraft shapes, we hope that we take one step closer to democratizing 3D content creation.

## 6 Broad Impact

Many fields such as AR/VR, robotics, architecture, gaming and film rely on high-quality 3D content. Creating such content, however, requires human experts, i.e., experienced artists, and a significant amount of development time. In contrast, platforms like Minecraft enable millions of users around the world to carve out coarse shapes with simple blocks. Our work aims at creating A.I. tools that would enable even novice users to upscale simple, low-resolution shapes into high resolution, beautiful 3D content. Our method currently focuses on 3D animal shapes. We are not currently aware of and do not foresee nefarious use cases of our method.

## 7 Disclosure of Funding

This work was funded by NVIDIA. Tianchang Shen and Jun Gao acknowledge additional revenue in the form of student scholarships from University of Toronto and the Vector Institute, which are not in direct support of this work.

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
