# OpenReview forum: "Deep Marching Tetrahedra: a Hybrid Representation for High-Resolution 3D Shape Synthesis"
_NeurIPS.cc/2021/Conference — NeurIPS 2021 Poster_

### Official Review · Reviewer_ztsv · 2021-07-13

**Rating:** 6
**Confidence:** 4

**Summary:**

This paper proposes deformable tetrahedral grid for 3D reconstruction. It aims to combine the advantage of both implicit surface and explicit surface by extracting the mesh using a differentiable marching tetrahedra layer and refine it using a grach convolutional neural network. The proposed method is evaluated an animal dataset collrected by the authors and ShapeNet objects, taking either a low-res voxel grid or a point cloud as input.

**Limitations And Societal Impact:**

The authors provide failure cases in the suppmat and discuss the broader impact of their work.

**Main Review:**

### Strengths
1. The paper is well-written and easy to follow.
1. It is interesting to define an adversarial loss on local regions for refining geometry.
1. The authors motivate the choice of MT instead of MC using a toy example, which I found interesting and convincing.
1. The proposed method demonstrates superior performance on 3D reconstruction of single objects, in particular for thin objects.
1. User studies are conducted to evaluate the quality of the 3D reconstruction of the animal dataset.

### Weaknesses

1. The proposed method is fairly similar to DefTet. As I understood, the major differences compared to DefTet are: a) using SDF instead of occupancy, b) hierarchical sampling, c) using GCN to predict vertice offsets and d) adversarial loss. Despite achieving better performance, these improvements appear to be incremental conceptually. Moreover, DefTet conducts experiments on Novel View Synthesis and Multi-view 3D Reconstruction, while the proposed method is only evaluated on 3D single object reconstruction from 3D input. What is the reason for this? It would be great if the author can elaborate more about the key differences compared to DefTet in the paper.

1. Considering the differentiability of MT/MC, I believe that it is also related to the loss. If there is a direct loss defined on the SDF field ($L_{sdf}$ as in this paper), I agree with the authors that the original MT/MC can be applied directly. However, it is not straightforward for me how the topology can be changed freely when only chamfer distance loss ($L_{cd}$) is considered. For example, it might be that $L_{cd}$ is not monotonically decreasing when transitioning from the initial topology to the target topology. Note that this is not a problem when using $L_{sdf}.Some more specific suggestions/questions regarding these concerns:

    a. In the main experiments $L_{sdf}$ is always applied. But the experiment of Fig.A in suppmat uses $L_{cd}$ only. While it is interesting, it is not straightforward to me how the original MT changes topology. It would be great if the authors can demonstrate the topology change of one single tetrahedron when using only $L_{cd}$. For example,  one can show how the topology changes from the left topology to the middle one, and from the middle to the right in Fig.3.

    b. The authors also argue that the refinement step using the GCN module is a reason that allows for topology change (L180). As I understood the topology change depends on the supervision signal (and hence the gradient) instead of the network architecture. It would be great if the authors can elaborate a bit more about this.

1. The idea of using an MLP to predict the implicit surface and extracting the explicit surface such that the loss can be directly applied on the surface is conceptually similar to MeshSDF [38]. It would be good if the author can explain why a comparison to MeshSDF is not considered.

1. While it is great that the authors provide ablation studies to verify some of the design choices, I found the ablation of some losses is missing (e.g. $L_{sdf}$ and $L_{def}).

1. The authors only consider experiments on single objects. It is unclear whether the method can be applied to room level (like in ConvONet).

1. This is not a major concern but the training scheme appears to be quite complicated with multiple steps and takes a very long time (6 days on 4 NVIDIA V100 GPUs).

In summary, I found this paper well-written and interesting but I have some major concerns regarding comparisons to existing works and the differentiability of MT/MC. I would like to increase my rating if the authors can address my concerns.

### Update after rebuttal

The authors have addressed my concerns regarding the differentiability of MT/MC. Therefore I vote for accepting this paper.

### Detailed comments

1. It would be good if the authors can provide the motivation for choosing the LSGAN objective instead of other GAN objectives.

1. Some of the citations are not up-to-date. For example, [6] and [37] are both published at conferences while they are still referred to Arxiv.


**Time Spent Reviewing:**

4

---

> ### Author Response · Authors · 2021-08-10
> **Authors' Response**
>
> We thank the reviewer for feedback and detailed comments. Below is our response to the concerns:
>
> [Differences Compared to DefTet] The major difference comparing with DefTet[16] is we have a differentiable marching tetrahedra layer to extract the surface, which allows joint training of topology and geometry, while the surface loss and occupancy loss in DefTet can not jointly optimize [16] (LN 90-95). Further improvements includes hierarchical sampling, surface subdivision, and adversarial loss, as mentioned by the reviewer.
>
> [Multi-view Reconstruction] We do not include experiments on tasks with image supervision due to time constraint. After submission, we found DMTET can be directly plugged into a current differentiable renderer [4] to jointly optimize the geometry, topology and color of surface mesh when supervised only by image loss, we provide an example here: https://dmtet.s3.us-west-1.amazonaws.com/spot_diff_render.mp4. Unlike DefTet [16], our method does not require reformulation of the diff-render pipeline. Note that this also demonstrates optimization of topology through MT from randomly initialized shape.
>
> [Typological change] We apologize for the confusions related to the topological changes. We provide one visualization https://dmtet.s3.us-west-1.amazonaws.com/bunny.mp4 where we optimize Chamfer Distance without L_sdf in 2D for easy elaboration. The vertex is colored based on the associated SDF value from negative (red) to positive (green). When the target surface is outside the current surface tetrahedra, the sign of SDF on the vertex can be flipped to change mesh topology. Similar results can also be obtained in 3D. We provided more analysis at Sec. A in Supplementary.
>
> [$L_{sdf}$ vs $L_{cd}$] We agree with the reviewer that $L_{cd}$ is not monotonically decreasing during optimization, so it may be stuck at local minimum. In practice, when training a network for a learning task, we found a combination of $L_{cd}$ with a less-weighted $L_{sdf}$ gives the best result. Our main contribution is that we demonstrate the additional supervision (e.g. chamfer distance, adversarial loss) defined directly on the extracted surface improves the shape synthesis quality, which is enabled by MT. The effect of each loss is clearly shown in ablation study:
>
>    • We compare the model trained with occupancy or SDF loss only with the model trained with additional surface based loss. We showed adding surface based losses captures better geometric details. (see Tab. 3, Tab. A, Fig. 7, and Fig. E).
>
>    • Our discriminator takes local SDF computed from explicit surface as input. The loss is propagated to the network through the MT layer, which results in typological changes that better model the details. (e.g. second row Fig. F and Tab. A.)
>
> [refinement step using the GCN] We agree with the reviewer that the capability of changing topology through MT is not related to the use of GCN and apologize for the confusion. We would like to clarify that in LN180 we mean GCN can refine the topology of initial prediction during inference, not the reason why MT can change topology through optimization.
>
> [Comparison with MeshSDF] We do provide comparison with MeshSDF in their experiment settings in Supplement Section A.
>
> [Scene Reconstruction] Our model can be applied for room-level scene reconstruction. We will include results in our revision.
>
> [Training Time] The major bottleneck during training is the computation of distance-based metrics, such as Chamfer Distance and Signed Distance, which requires computing point-to-point and point-to-face distance for millions of pairs. Currently in our implementation we compute the distance on the fly. It is possible to speed up training by using e.g. pre-computed SDF values or sampling less points when computing CD (currently we sample 100k points).

---

> ### Author Response · Authors · 2021-08-29
> **Following up regarding the discussion**
>
> Dear reviewer ztsv,
>
> We believe we have addressed your concerns and clarified the confusion in the previous reply. Since the discussion period is ending, we kindly ask if there is any updated assessment or concern of our paper. Thanks for your consideration!

---

> > ### Comment · Reviewer_ztsv · 2021-08-29
> > **Remaining concerns**
> >
> > Thanks for the responses and the new results. I have a few remaining concerns. The authors agree that the loss of the MT layer is not monotonically decreasing during optimization and might be stuck at local minimum. Therefore the direct supervision of the SDF is still important. It is unfortunate that the authors do not provide results without $L_{sdf}$ under the learning setting. In my opinion, one of the major contributions of this paper is that the original MC/MT can be applied in a learning framework. Thus it appears to be important to understand when it works and when not (it is not surprising that it works with $L_{sdf}$). In addition, I slightly agree with other reviewers regarding the limited technical novelty. These concerns make me currently stay at a borderline recommendation (very slightly towards negative).
> >
> > More thoughts/suggestions on the writing:
> > - I agree with reviewer QWSU that the "proposed" MT layer is the original MT algorithm, and it might be slightly confusing to call it "differentiable MT".
> > - The authors' reply to reviewer QWSU that *"We wish to clarify that we do not claim that solely using supervision on the surface extracted by MT can produce the best result in all cases."*. I think this misunderstanding might be caused by the introduction, which indicates that the proposed method allows for directly optimizes the "surface mesh". Thus the readers did not expect supervision on both implicit and explicit surface following the story of the introduction.

---

> > > ### Author Response · Authors · 2021-08-30
> > > **Authors' Reponse**
> > >
> > > We thank the reviewer for the updated comments! We are surprised to see that the use of $L_{sdf}$ becomes a general concern, which is a common practice in prior works that couple explicit supervision with implicit fields [38, A1, A2]. We thank the reviewer for pointing out the sentence in our paper that leads to this concern, and we will make our statement clear in a revision. Moreover, we argue that the problem of local minimum depends on the choice of loss function on the surface, and it is not introduced by MT/MC. More specifically,
> > > * **Chamfer distance** itself has local minimum s.t. a surface point can be matched to two points at opposite directions in Eqn. 3. This is independent to how the surface is parameterized. Prior works on explicit representation training on CD need carefully designed regularization losses to avoid bad geometry even when the mesh topology is fixed and matches the GT (i.e. both are genus-0 surface) [46, 18, 16]. In our case, we find there is no guarantee that the shape converges to GT from a random initialization (even overfitting), so we pair with $L_{sdf}$ in learning to alleviate this issue. Otherwise, the result is worse than solely using SDF loss (e.g with. double/broken surface). We stress that prior works on differentiable iso-surfacing [38, A1] also incorporate $L_{sdf}$ to achieve best result. We don't think our formulation is worse in this regard and compare with [38]. Again, the advantage of differentiable iso-surfacing techniques is to allow **additional** supervision/regularization on implicit field. Both [38] and our work shows improvement from it (CD and $L_{adv}$). In addition, we show topology changes smoothly when gradient on surface is consistent (animation in previous reply and Fig. A), thus we argue it is not a problem in the differentiability of MT.
> > > * The only related work, to our best knowledge, that does not use $L_{sdf}$ is [26]. However, they need a cubic prior and smoothness loss on occupancy field to ensure no double/broken surface is generated (see Fig. 5 in their paper). We find adding regularization does help our method converge to correct geometry in overfitting cases. However, it also suppresses reconstructing high-frequency details, which is an undesirable side effect for high-resolution shape synthesis tasks. Since we have GT shapes in our tasks, we use a less-weighted $L_{sdf}$ loss instead. The improvement from surface supervision is carefully ablated in our experiments and user studies, which we believe justifies our contribution. We also compare with [26], which is limited to low resolution in learning due to the computation of expectation over all possible configurations in MC.)
> > > * We observe that the problem of local minimum is less a concern for image supervision. In practice, DMTET converges to correct topology from randomly initialized signed distance field without $L_{sdf}$ (see animation in the previous reply), which is hardly the case for CD.
> > > In summary, we argue that the local minimum issue of CD is not specific to our formulation of differentiable iso-surfacing layer. However, we agree that the discussion and the ablation should be added to our paper for completeness. We will include it in a revision with failure cases.
> > >
> > > **[Technical Novelty]** We are sorry to hear we didn’t address your concern. We wish to briefly stress a few points:
> > >
> > > We kindly disagree that “it is not surprising that it (MT) works with L_sdf”. It has been a common misunderstanding in the community ([26, 38] and in the rebuttal discussion) that the singularity in MT/MC prohibits topological change. We address this and introduce a carefully designed coarse-to-fine strategy specific to MT. None of the techniques we leveraged (MT, volume/surface subdivision) has been explored in learning set-up, to our best knowledge. It is the clever design of putting them together, and making them end-to-end trainable, forms the main contribution and success of DMTET. We believe a simple formulation of differentiable iso-surfacing that scales to high resolution is important for future work to explore explicit supervision on neural implicit representation, maybe a better surface loss that alleviates the local minimum issue.
> > >
> > > **[Differentiable MT]** We will emphasize that we use MT as a differentiable layer in a revision. Thank you for pointing it out!
> > > Additional References:
> > >
> > > [A1] Iso-Points: Optimizing Neural Implicit Surfaces with Hybrid Representations, Wang et al. 2021
> > >
> > > [A2] Coupling Explicit and Implicit Surface Representations for Generative 3D Modeling, Poursaeed et al. 2020

---

> > > > ### Comment · Reviewer_ztsv · 2021-09-01
> > > > **increased score**
> > > >
> > > > Thanks for the further clarifications. I played with some 2D toy examples and I am convinced by the argument that the loss has local minima but it is not introduced by MT/MC. Therefore I will increase my score and vote for accepting this paper.
> > > >
> > > > Regarding my argument "it is not surprising that it (MT) works with L_sdf", I still believe it is true to some extent as L_sdf defined on the *implicit surface* alone allows for learning arbitrary topology, regardless of whether an additional loss is defined on the *explicit surface*. This means L_sdf can take care of the topology change while the loss defined on the explicit surface (L_cd in this paper) only needs to take care of small variants within a fixed topology. In this case, one can also directly use the original MT/MC even if it does not allows for changing the topology freely.

---

### Official Review · Reviewer_mF8Y · 2021-07-14

**Rating:** 6
**Confidence:** 4

**Summary:**

This paper presents a method to generate fine-grained shape from coarse volume or sparse point clouds. The pipeline starts from a feature extractor that produces a feature volume from the input, which is then used to predict and refine sdf values on a pre-defined  tetrahedral grid. Both the vertex location and sdf are then updated, with surface subdivision to capture more local features. At last, a differentiable marching tetrahedra is run to get the mesh. The model is supervised by both Chamfer distance on points sampled from the surface, as well as the SDF of tetrahedra vertices. The overall performance on shape synthesis with coarse volume or point cloud achieves the best quantitative performance compared to the baselines, and the advantages of MT over marching cube are analyzed.

**Limitations And Societal Impact:**

No comment.

**Main Review:**

Strength:
- The use of deformable tetrahedra seems to be an elegant, i.e. accurate and efficient, representation for 3D shapes.
- The performance on both tasks clearly outperform baseline methods, and the qualitative results looks promising.

Weakness:
- The task achieved by this paper is 3D super-resolution rather than shape synthesis. The pipeline creates a one-to-one mapping between the input and output, and therefore I don't see how novel shape could be synthesized.
- The deformable tetrahedra is a generic representation not only just valid for the proposed tasks. To fully demonstrate the effectiveness, other common tasks, e.g. shape from image, reconstruction, could be investigated.
- It might be good to have some tight comparison to the marching cube based pipeline under two tasks. I see both of the tasks could be similarly achieved with regular cubical grid plus deep marching cube [26]. It is good that DMC [26] is compared in table 3. However, one might be interested to see how the pipeline work with exactly the same pipeline using cube instead of tetrahedra. In this way, we would be able to clearly see the advantage of using tetrahedra.
- When compare MT and MC, run-time efficiency should be provided.

Overall, this work provides an extension of the marching cube based approach with tetrahedra. The major novelty seems to be more engineering oriented (if I didn't miss anything important), and adding more experiments might be helpful to highlight the importance of the contribution.

**Time Spent Reviewing:**

1.5

---

> ### Author Response · Authors · 2021-08-10
> **Authors' Response**
>
> We thank the reviewer for feedback and detailed comments. Below is our response to the concerns:
>
> [Technical contribution] We wish to clarify that our model is not engineering-oriented, instead, it carefully considers the problems in the current 3D literature and proposes novel solutions to it. We summarize our technical contributions below:
> 1)  In contrast to the analysis in (DMC [26], MeshSDF[38]), we show that using marching tetrahedra (MT) as a differentiable iso-surfacing layer allows topological change, and compare with [26, 38]. To our best knowledge, we are the first to introduce MT as a differentiable module in a DL framework, and demonstrate that the additional supervision (e.g. chamfer distance, adversarial loss) defined directly on the extracted surface improves the shape synthesis quality.
> 2) The original MT (or MC) algorithm has a cubically growing computation cost and thus does not scale up to high-resolution in a training setup. We introduce a coarse-to-fine optimization strategy that achieves better reconstruction quality than state-of-the-art methods, while requiring a lower computation cost.
>
> [Synthesizing novel shapes] Although our model learns 1-1 mapping, it can generate novel shapes by changing the input voxels. In Fig. 5 we compare the synthesized shape with the closest retrieved shape from the training set to demonstrate that novel shapes are generated for unseen inputs. Moreover, our model generalizes to artist-created shapes despite the domain gap, as shown in Fig. 6.
>
> [Applying DMTET to other tasks] Thanks for the suggestion. We apply our proposed representation to point cloud 3D reconstruction and voxel super-resolution tasks to demonstrate DMTET as a generic 3D representation. We also found the mesh generated by DMTET can be directly fed into diff-renderer [4] for reconstruction from Multi-view images. In contrast to DEFTET, DMTET does not require modification to the current diff-rendering pipeline. The include visualization here: https://dmtet.s3.us-west-1.amazonaws.com/spot_diff_render.mp4. We will add it in a revision.
>
> [Comparing MT with MC] Thanks for pointing this out. We compared MT with MC in a non-learning setup in Sec. 4.2.1, and MT achieved better performance than MC. We will add the suggested experiment comparing MC and MT in the learning setup in our revision.

---

### Official Review · Reviewer_QWSU · 2021-07-15

**Rating:** 5
**Confidence:** 5

**Summary:**

This paper presents a method for synthesizing 3D shapes from coarse voxels using a 3D conditional generative model.
The main *claimed* contribution is a differentiable marching tetrahedra layer that can convert the implicit signed distance field to the explicit surface representation. The paper also claims that with such a differentiable marching tetrahedra (MT) layer, one can jointly optimize the surface geometry and topology during training.
Despite the fact that the presented results do look promising compared to previous works, I doubt that the improvement comes from the proposed differentiable MT layer, due to the fundamental technical flaws which I will elaborate on later.
Instead, I believe a great part of the performance comes from the direct and full SDF supervision provided in the seemingly minor *regularization* term and the deformable tetrahedral grid, which the authors fail to provide ablation study on these two algorithmic components.


**Limitations And Societal Impact:**

Yes.

**Main Review:**

- The technical contribution of the proposed work is very limited. The paper claims that the main contribution comes from the combination of the deformable tetrahedral grid and the differentiable marching tetrahedra (MT) layer. Since the deformable tetrahedral grid is an existing technique, the only new algorithmic component is the differentiable MT layer. However, in this work, they actually directly apply the traditional MT algorithm for converting the signed distance field to mesh representation, with *NO* modification. Though the work also incorporates volume and surface subdivision techniques, these are very common approaches for coarse-to-fine reconstruction.

- There is a fundamental flaw in claiming the original marching tetrahedra algorithm is differentiable. As agreed by the authors, there is a singularity in the original MT algorithm -- when the signs of the edge endpoints are the same, there are no gradients flowing into this edge. In addition, in line 218 to 220, the paper also admits that only the vertices that are close to the iso-surface in the tetrahedral grid receive gradients, while the other vertices do not. Hence, I do not think it is fair to call the proposed MT layer (actually the original MT algorithm) differentiable.

- The writing is very misleading, as it tries to convince the readers that the main improvement comes from the differentiable MT layers (the main contribution of the paper). However, as analyzed in the above point, due to the fundamental flaw in the differentiability of the MT layer, it is very unlikely (if not impossible) that solely using the proposed MT layer can achieve a large topology change to generate high-fidelity results. Hence, I do not believe the proposed MT layer contributes a great part in achieving good performance in the results. Instead, the performance boost should probably be credited to the deformable tetrahedral gird (an existing technique) and the full SDF supervision in the so-called *regularization* term. However, the paper fails to provide any ablation study regarding the effect of these two components.
I would like to see the performance without the SDF regularization term, the deformable tetrahedral grid, and both. Otherwise, I do not think the evaluation is comprehensive.

**Needs Ethics Review:**

Yes

**Time Spent Reviewing:**

4

---

> ### Author Response · Authors · 2021-08-10
> **Authors' Response**
>
> We thank the reviewer for the detailed comments. We address them below.
>
> First, the reviewer seems to have misunderstood our argument about singularity and differentiability - we clarify it here: in practise, the equation (Eq. 1 in Suppl) of MT layer is only evaluated when the signs of the edge endpoints are **different** $\text{sign}(s(v_{b}))\neq \text{sign}(s(v_{a}))$. Therefore, the mentioned singularity ($\text{sign}(s(v_{b})) = \text{sign}(s(v_{a}))$) will never appear in the practise, and the gradient can flow back to both, the SDF value and positions of the vertices in the tetrahedral grid (LN147-151 in the main paper, and Sec. A in Supplementary material). Thus, we call the proposed MT layer differentiable. We can adjust the claim if the reviewer still has concerns.
>
> We agree that the gradient can only flow to vertices around the surface, but learning from the surface loss through MT can achieve large topological changes by gradually deforming the underlying surface. We provide a demonstration in Fig. A in Supplementary to show the topological changes when optimizing chamfer distance **without** L_sdf. We further provide a 2D example here https://dmtet.s3.us-west-1.amazonaws.com/bunny.mp4 where we optimize Chamfer Distance **without** L_sdf (GT points are in purple and the predicted surface is in red). We can still see that the optimized results are close to the target and have very different topology than the initialization.
>
> We further wish to clarify our technical contributions and our experiments that consolidate the contributions below:
> 1)  In contrast to the analysis in (DMC [26], MeshSDF[38]), we show that using marching tetrahedra (MT) as a differentiable iso-surfacing layer allows topological change, and compared with [26, 38]. To our best knowledge, we are the first to introduce MT as a differentiable module in a DL framework, and demonstrate that the additional supervision defined directly on the extracted surface improves the shape synthesis quality as demonstrated in the following experiments:
>
>       •	In ablation study, we compare the model trained with occupancy or SDF loss with the model trained with additional surface loss (see Tab. 3, Tab. A, Fig. 7, and Fig. E).
>
>       • Our discriminator takes local SDF computed from an explicit surface as input. The loss is propagated to the network through the MT layer, which results in typological changes and gets much better details. (e.g. second row Fig. F)
>
> 2) The original MT (or MC) algorithm has a cubically growing computation cost and thus does not scale up to high-resolution in a training setup. We introduce a coarse-to-fine optimization strategy that achieves better reconstruction quality than state-of-the-art methods, while requiring a lower computation cost.  Our method is composed of modules that extend existing work to allow end-to-end training and address the known limitations (discussed in 3.2.1).

---

> > ### Comment · Reviewer_QWSU · 2021-08-27
> > **Major concerns still exist**
> >
> > Thanks for the response and the demo!
> > - After reading the rebuttal, now I agree that the MT operation is differentiable by itself in practice if it is only evaluated when the signs of the edge endpoints are different. Then this brings another question: what is the novel contribution of the paper when the MT algorithm is differentiable by itself? Implementing a differentiable module in the deep learning framework is more like an engineering task. The deformable tetrahedral grid is an existing technique. Though the coarse-to-fine optimization is a nice call, it is a widely used methodology in many applications and thus cannot be counted as a NeurIPS-level contribution.
> >
> > - Besides, I am a bit disappointed not to see any quantitative ablation study regarding the $L_{SDF}$. The demo is really nice. However, without a quantitative result, it is difficult to determine 1) whether the MT layer alone can achieve significant results in all cases; 2) how does  $L_{SDF}$ affect the final results. Without such an evaluation, I am still not convinced that the MT layer only is able to handle all kinds of difficult local minima.
> >
> > I am happy to hear from the authors if I miss something.

---

> > > ### Author Response · Authors · 2021-08-28
> > > **Authors' Response**
> > >
> > > We thank the reviewer for the reply! We are glad to hear that we addressed the main concern regarding the differentiability of MT. Here we would like to tackle the remaining concerns:
> > >
> > > [Novelty] We argue that not only new methodology but also analysis that provides new views of existing techniques, should be considered as novel contributions. More specifically,
> > > * In contrast to the common misunderstanding that the singularity in formulation prohibits topological change [26, 38] in MT, we demonstrate MT is differentiable and allows topological change. This lets us have a much more simple and efficient technique that does not require computing expectation of all configurations [26] or performing an additional forward pass [36]. We think our analysis is valuable for the 3D community.
> > > * Prior works on differentiable iso-surfacing [26, 38] only show improvements in **optimization or low-resolution** cases due to scalability issues. With deformable grid and coarse-to-fine optimization, we scale our method to high resolution and are the first to demonstrate the improvement from surface supervision (Chamfer Distance and Adversarial Loss) on large-scale 3D synthesis tasks, which we believe is an exciting result!
> > > * It is a known limitation of MT/MC that it cannot capture topology features smaller than the grid resolution. Optimizing the surface via MT makes the implicit representation aware of the surface extraction step, thus alleviating this limitation (Sec. 4.2.1).
> > > * We show that MT is more performant than MC for reconstructing the surfaces from discretized implicit fields (Sec. 4.2.1). Combined with the deformable tetrahedral grid, which brings additional gains, is another advantage, but we don’t claim this as contribution on its own.
> > >
> > >
> > > In summary, we believe that our analysis and experimental results provide a novel view of widely-adopted techniques in 3D DL context. Introducing a simple and efficient differentiable iso-surfacing layer can enable new methods for tasks that favor or require explicit supervision (e.g. [Multi-view Reconstruction] in our reply to reviewer ztsv). The limitations addressed and improvement made by DMTET over SOTA methods should not be neglected only because we don’t modify the formulation of MT. We kindly invite the reviewer to view our work in a more holistic manner.
> > >
> > > [$L_{SDF}$] We wish to clarify that we do not claim that solely using supervision on the surface extracted by MT can produce the best result in all cases. Here we refer to our reply to reviewer ztsv:
> > > > We agree with the reviewer that $L_{cd}$ is not monotonically decreasing during optimization, so it may be stuck at local minimum. In practice, when training a network for a learning task, we found a combination of $L_{cd}$ with a less-weighted Lsdf gives the best result. Our main contribution is that we demonstrate the additional supervision (e.g. chamfer distance, adversarial loss) defined directly on the extracted surface improves the shape synthesis quality, which is enabled by MT. The effect of each loss is clearly shown in ablation study:
> > > > * We compare the model trained with occupancy or SDF loss only with the model trained with additional surface based loss. We showed adding surface-based losses captures better geometric details. (see Tab. 3, Tab. A, Fig. 7, and Fig. E).
> > > > * Our discriminator takes local SDF computed from the explicit surface as input. The loss is propagated to the network through the MT layer, which results in typological changes that better model the details. (e.g. second row Fig. F and Tab. A.)
> > >
> > > Moreover, we think the problem of local minimum is a general limitation of optimizing the surface loss which is not monotonically decreasing w.r.t. surface position, not specific to MT. Prior works on neural implicit surfaces [38 and “Iso-Points: Optimizing Neural Implicit Surfaces with Hybrid Representations”, Yifan et al.] also leverage both supervision on implicit fields and surfaces to achieve better results than solely using the former. Our work follows the same motivation.

---

> > > > ### Comment · Reviewer_QWSU · 2021-08-28
> > > > **My final rating**
> > > >
> > > > Thanks for the new response! However, it still cannot convince me that the contribution is sufficient enough for NeurIPS. In addition, the lack of ablation study still remains. But considering the good results and practical meaning of the technique, I raise my score to marginally below the acceptance threshold.

---

### Official Review · Reviewer_MFHh · 2021-07-16

**Rating:** 7
**Confidence:** 4

**Summary:**

The authors propose an approach (DMTet) to obtain detailed 3D shape from coarse voxels/pointclouds.
Given an input DMTet predicts SDF values at discrete tetrahedral grid. Authors use this SDF to obtain a coarse surface which is further used by GCN to refine the tetrahedral grid. Surface is then recomputed. Authors use a second GCN to predict parameters for loop subdivision to further increase the mesh resolution.


**Limitations And Societal Impact:**

Authors discuss limitations of their method and broader impact in the paper.

**Main Review:**

+ The demonstrated results look good.
+ Authors include detailed evaluations in supplementary which are appreciated.
+ Paper is well written and easy to follow.

A few comments:
1. Supplementary, Sec E.2 Table B. Why do the numbers for point cloud reconstruction on ShapeNet for baselines look different than Table 1 in [16]? Was the experimental setup somehow different?
2. Eq. 1,2: f(v_i) is input to the GCN, what is the advantage of reconstructing it back?
3. L286-296: Numbers could be put in a table for easier parsing.
4. Fig 7: I find it funny that the lamp shown in the second row to showcase superior performance of the method is also used in supplementary fig. J to show the limitations of the method.

Animals and shapenet objects are relatively smooth shapes. Can the proposed method be used to generate detailed shapes such as clothed 3D humans? This would be a challenging testbed to assess the gains from the proposed surface refinements (adversarial, volumetric and surface).

**Time Spent Reviewing:**

7

---

> ### Author Response · Authors · 2021-08-10
> **Authors' Response**
>
> We thank the reviewer for positive feedback and detailed comments. Below is our response.
>
> [Differences with numbers reported in [16]]: We reproduced the methods in [16] and got better performance than the original [16]. All methods use the same PCD encoder and are evaluated under the same pipeline to ensure a fair comparison.
>
> [f(v_i)] The f(v_i) that is output by the GCN is the updated (not reconstructed) feature for each vertex, which is the input to the next surface refinement module. We will denote it with a different symbol in the revision to avoid confusion. Thanks for pointing it out.
>
> [Testing on clothed 3D humans] We are working on extending our method to more object categories. We thank the reviewer for the suggestion.

---

### Review · Ethics_Reviewer_NXfi · 2021-08-09

**Recommendation:** N/A

**Ethics Review:**

The issues raised by the reviewers are related to the technical quality of the work and scholarship rather than ethical considerations with societal implications.

---

### Review · Ethics_Reviewer_CRaS · 2021-08-10

**Recommendation:** N/A

**Ethics Review:**

The paper aims to optimize the creation of high quality 3D contents and experiment on data of animals. The designed domains of the technique are AR/VR, robotics, architecture, gaming and film, etc. Therefore, it seems to me that there is no issues raised by the technique as long as it's applied to the target domains non-related to people.

---

### Decision · Program_Chairs · 2021-09-28

**Decision:**

Accept (Poster)

**Comment:**

The paper initially received mixed reviews. After the rebuttal, three reviewers were positive while one remained negative. The shared concerns include the limited technical contribution, the missing ablation studies on $L_{sdf}$, and the lack of a comprehensive evaluation between MT and MC. The AC shares these concerns.

But the AC also agrees with the reviewers that all engineering efforts in making the pipeline working and achieving good results, as well as the potential impact to the community on providing an easy solution to directly adopting MT/MC in a neural network, are useful contributions. The AC, therefore, recommends acceptance of this borderline case. The authors are strongly encouraged to include the missing ablation studies and comparisons with MC in the camera ready.

**Consistency Experiment:**

NeurIPS has a long history of experimentation. In 2014, NeurIPS ran an experiment in which 10% of submissions were reviewed by two independent committees to quantify the randomness in the review process. This year, we repeated a variant of this experiment to see how the quality of the review process has changed over time.  This paper was part of the experiment and was therefore assigned to two committees (consisting of reviewers, an Area Chair, and a Senior Area Chair) that reached independent decisions.  If both committees made the same recommendation, this recommendation was followed. If a single committee recommended acceptance, the paper was accepted (with the exception of a few cases in which the other committee identified what we considered a fatal flaw, e.g., an error in a key result).

Both committees reached the same decision: **Accept (Poster)**

The other committee assigned to the paper recommended **Accept (Poster)**.  You can find the other set of reviews, along with any follow up discussion with the authors here:
https://openreview.net/forum?id=xN3XX6pKSD5